# Impact on Isoleucine and Valine Supplementation When Decreasing Use of Medical Food in the Nutritional Management of Methylmalonic Acidemia

**DOI:** 10.3390/nu12020473

**Published:** 2020-02-13

**Authors:** Laurie E. Bernstein, Casey Burns, Morgan Drumm, Sommer Gaughan, Melissa Sailer, Peter R. Baker II

**Affiliations:** Department of Pediatrics Section of Clinical Genetics and Metabolism, Aurora, Children’s Hospital Colorado, University of Colorado, Anschutz Medical Campus, Aurora, CO 80045, USA; laurie.bernstein@childrenscolorado.org (L.E.B.); morgan.drumm@childrenscolorado.org (M.D.);

**Keywords:** methylmalonic acidemia (MMA), methylmalonyl-CoA mutase deficiency, medical food, leucine, isoleucine, valine

## Abstract

Background: Methylmalonic acidemia (MMA) is an autosomal recessive disorder treated with precursor-free medical food while limiting natural protein. This retrospective chart review was to determine if there was a relationship between medical food, valine (VAL) and/or isoleucine (ILE) supplementation, total protein intake, and plasma amino acid profiles. Methods: A chart review, of patients aged 31 days or older with MMA treated with dietary intervention and supplementation of VAL and/or ILE and followed at the Children’s Hospital Colorado Inherited Metabolic Diseases Clinic. Dietary prescriptions and plasma amino acid concentrations were obtained at multiple time points. Results: Baseline mean total protein intake for five patients was 198% of Recommended Dietary Allowance (RDA) with 107% natural protein and 91% medical food. Following intervention, total protein intake (*p* = 0.0357), protein from medical food (*p* = 0.0142), and leucine (LEU) from medical food (*p* = 0.0276) were lower, with no significant change in natural protein intake (*p* = 0.2036). At baseline, 80% of patients received VAL supplementation and 100% received ILE supplementation. After intervention, only one of the cohort remained on supplementation. There was no statistically significant difference in plasma propiogenic amino acid concentrations. Conclusions: Decreased intake of LEU from medical food allowed for discontinuation of amino acid supplementation, while meeting the RDA for protein.

Concise 1-sentence Take Home Message

Decreased intake of LEU from medical food may allow for the discontinuation of amino acid supplementation and precursor-free medical food, while meeting the RDA for protein.

## 1. Introduction

Methylmalonic acidemia (MMA) is an autosomal recessive disorder, characterized by an enzymatic defect in propiogenic amino acid metabolism due to methylmalonyl-CoA mutase (*MUT*) deficiency, limiting the conversion of methylmalonyl-CoA to succinyl-CoA. The precursor amino acids that are catabolized in this pathway include isoleucine (ILE), valine (VAL), methionine (MET), and threonine (THR) [1]. Left untreated, concentrations of methylmalonic acid and/or related metabolites accumulate, contributing to vomiting, poor feeding, failure to thrive, dehydration, hypotonia, and severe metabolic acidosis, within the first few days of life. Most individuals with MMA can be detected through newborn screening using propionylcarnitine elevations, and treatment is initiated at that time. Lifelong therapy consists of carnitine supplementation and nutritional therapy. For some individuals, specifically Cobalamin defects, B12 injections are also therapeutic and nutritional therapy may not be necessary. The latter, historically, has been the primary means of chronic MMA management; however, their efficacy and safety remain unclear as high incidents of morbidity and mortality continue to be observed in this population [2]. Dietary treatment consists of a precursor propiogenic amino acid restricted diet, while providing adequate intake of essential propiogenic amino acids and supplementing the remainder of age- and sex-appropriate protein needs with precursor-free medical food [1,2,3,4,5]. Despite dietary treatment, low plasma levels of VAL and ILE typically occur, prompting the supplementation of these individual amino acids to prevent VAL and ILE deficiencies [2].

Historically, nutrition management regimens for MMA have included a natural protein restriction, precursor-free medical food, and VAL and/or ILE supplementation [3]. Recently, this approach has been investigated, based on the finding that natural protein restriction (or over-restriction) and high intake of precursor-free medical food skew plasma amino acid profiles, largely due to the high leucine (LEU) content in precursor-free medical foods [2,6]. This potentially places the patient at risk for iatrogenic problems of amino acid transport and metabolism [1].

This study collected retrospective data on diet changes implemented based on conclusions drawn from the presentation: *Methylmalonic Acidemia—Current Dietary Treatment and a Different Approach to Management* presented by Irini Manoli, MD, PhD, from the Organic Acidemia Research Section of the Human Genome Research Institute at the National Institute of Health, at the Broadening Horizons in Clinical Practice 5th GMDI Educational Conference 2014, and the subsequently published data by Manoli et al. [2]. There was a clinical decision to decrease use of precursor-free medical food and VAL AND/OR ILE supplementation in the management of children and adults with MMA due to MUT deficiency. The primary goal of this retrospective study was to determine the relationship between protein/LEU from precursor-free medical food consumption, VAL and/or ILE supplementation, total protein intake, and amino acid profiles evaluated before and after dietary management changes. We present dietary efficacy data from a single clinic, and validate findings from the NIH. 

## 2. Materials and Methods

This study was a retrospective chart review carried out at the University of Colorado/Children’s Hospital Colorado Inherited Metabolic Diseases clinic. EPIC (electronic medical record) was accessed to obtain historical diet prescriptions; patient laboratory values; and composition of protein prescribed, including total protein, natural protein, and protein and LEU from precursor-free medical food. Protein values were calculated per kilogram of body weight and compared to appropriate age and sex established Recommended Dietary Allowances (RDA). RDA for protein intake was based on the Institute of Medicine Dietary Reference Intakes for protein and amino acids [7]. Plasma amino acid profiles reflecting the dietary prescription were collected sequentially, in conjunction with change in dietary management, in a fasting state (more than 3 h following last oral intake), as per our standard of clinical practice. Amino acids were analyzed in Children’s Hospital Colorado Biochemical Laboratory using standard methods and protocols on the Biochrom 30 Amino Acid Analyzer (Cambridge, UK). Data were extracted from five patients with MMA from January 2012 until October 2017. Inclusion criteria were children aged 31 days or older and adults diagnosed with MMA (caused by MUT deficiency), who were treated with dietary intervention with VAL and/or ILE supplementation and followed for management at the University of Colorado. Exclusion criteria included children and adults with other organic acidemias (including other etiologies of MMA such as cobalamin metabolism defects and B12 responders), children and adults with MMA with lack of laboratory follow up, and patients who are not regularly followed at the University of Colorado.

Paired Student *t*-test, Chi square analysis, and Metabolic Pro were used to analyze differences in dietary composition, dietary amino acid content, and plasma amino acid concentrations before and after dietary changes that decreased protein from medical food and maximize intact/natural protein. Two-tailed *p*-value of <0.05 was considered statistically significant. Statistical analysis and graphics were completed in Prism 7.0 Graph Pad Software Inc. (La Jolla, California USA,). University of Colorado COMIRB #17-1236.

## 3. Results

At the beginning of the data collection period, five patients with mut^0^ MMA phenotype (five male non-B12 responsive, one of which was post liver transplant; mean age 11.8 years, range 6–18 years) were included. Upon completion of the study timeframe, mean age was 13.6 years with a range of 8–20 years. Diet changes occurred over a mean of 6.6 months (range = 0–15 months) with a mean number of 3.8 diet changes (range = 1–8 diet changes). Figure 1 and Table 1 illustrate changes in protein and LEU content of the prescribed diet in MMA patients over time. Mean baseline total protein intake for patients with MMA was 198% of the RDA prior to diet change [7]. Of this, natural protein intake represented 107% while protein from precursor-free medical food represented 91%. Most patients received multiple diet changes to slowly decrease total protein intake by decreasing protein from precursor-free medical food. After the final diet change, mean total protein intake was reduced to 141% of the RDA [7] (*p* = 0.044). Natural protein slightly increased to 123% (*p* = no significance (NS)), and protein from precursor-free medical food dramatically decreased to 18% (*p* = 0.014). LEU intake from precursor-free medical food correspondingly decreased from 125 mg/kg/d to 26 mg/kg/d (*p* = 0.028). Table 2 illustrates individual changes in total protein, natural protein, and protein from precursor-free medical food for each patient from baseline to final diet change. 

Need for supplementation with VAL and/or ILE, before and after dietary change was evaluated. At baseline, 100% of the cohort (*n* = 5) were prescribed ILE supplementation, with a mean dose of 550 mg/day. The majority of the cohort (*n* = 4, 80%) were prescribed VAL supplementation, with a mean dose of 50 mg/day. With subsequent decreases in medical food intake and decreasing mean total protein intake, only one patient continued supplementation after the completion date of the chart review. Furthermore, we noted a decrease in average LEU:VAL and LEU:ILE ratios from baseline to final diet change. At baseline LEU:VAL = 2.6:1 and LEU:ILE = 3.5:1 by the final diet change average ratios showed LEU:VAL = 2.0:1 (*p* = NS) and LEU:ILE = 3.0:1 (*p* = NS), respectively. While not significant, these decreased ratios are similar to previously reported patterns [2]. Table 3 illustrates quantitative propiogenic amino acid concentrations taken before and after change in diet composition. With changes in dietary protein intake and individual amino acid supplementation, there were no statistically significant changes in plasma amino acids (ILE, VAL, LEU, MET) with the exception of THR. Individual patient results demonstrating serum amino acid levels in relation to decreased supplementation of VAL and ILE while reducing LEU intake from precursor-free medical food are illustrated in Appendix A. 

## 4. Discussion

Leucine (LEU) is essential to growth and whole-body amino acid metabolism, arguably more than other amino acids. The importance of LEU in activating protein synthesis in skeletal muscle has been widely described [9,10]. Additionally, human tissue and food-derived proteins contain larger amounts of leucine than any other amino acid. Furthermore, the WHO/FAO/UNU recommendations for appropriate intake of amino acids indicate a higher need for LEU at 39 mg/kg/day than for any other amino acid with isoleucine at 20 mg/kg/d and valine at 26 mg/kg/day [8,9]. However, from the initial studies describing management of MMA, it has been known that excess LEU intake can decrease plasma levels of the other essential branched chain amino acids ILE and VAL [11]. LEU has biological effects not only in growth and protein accretion, but also in cell signaling, insulin sensitivity, central nervous system satiety sensing, and cellular energy metabolism, as has been recently well summarized by Myles et al. [6]. Clinical dietary management of propiogenic organic acidemias has largely focused on medical foods that are disproportionately high in LEU and provide total protein intake well above that recommended by the RDA. The safety and efficacy of such dietary therapy has only recently come under scrutiny [2,5,6].

Several recent papers, particularly Manoli et al. [2], support the idea that high intake of protein from precursor-free medical food containing LEU but little to no VAL or ILE has a negative effect on the plasma VAL and ILE levels [2,6]. Although the mechanism is not completely understood, it is thought that this is due to leucine having an inhibitory role on the branched-chain ketoacid dehydrogenase-kinase, which then results in increasing branched chain amino acids (BCAA) oxidation from increased activation of branched-chain ketoacid dehydrogenase [2]. In the largest and most recent study from the NIH, a total of *n* = 46 patients with MUT-deficient MMA were studied regarding dietary composition and intake. The initial hypothesis was that natural protein consumption was deficient relative to the intake of incomplete protein from precursor-free medical foods. Unexpectedly, despite natural protein intake meeting the RDA for many patients and total protein intake well beyond that recommended by the RDA, VAL and ILE were deficient and had to be supplemented. The investigators concluded that this was due to the increased amount of LEU (and consequently higher LEU:protein ratio) from precursor-free medical foods [2]. 

Propionic acidemia (PA) is a disease closely related, and managed similarly, to MMA. Recent GMDI guidelines for the nutritional management of PA supported the use of increasing natural protein rather than using VAL and/or ILE supplementation to correct VAL and/or ILE plasma levels [12,13]. They encourage use of propiogenic free medical food only when the RDA is not being met for intact protein. They further suggest limiting the additional medical food protein to meet 100–120% of the RDA needs when combined with intact protein, thereby limiting the LEU load from medical food [12,13]. Our data show that we provided over 100–120% of total protein, and suspect overprescribing of total protein has been a common approach to nutrition management. 

Our findings support those reported in Manoli et al. [2,6,12,13], and are in line with recent recommendations for dietary management of propiogenic organic acidemias. In our cohort, patients meeting or exceeding natural protein from the RDA, and receiving higher amounts of LEU from precursor-free medical foods, needed VAL and ILE supplementation prior to change in dietary management. After decreasing the amount of precursor-free medical food, LEU intake decreased significantly from baseline to the final diet change as precursor-free medical food protein was decreased. VAL and ILE supplementations were no longer necessary in the majority of our patients to maintain normal plasma levels. Plasma VAL and ILE levels, as well as other propiogenic amino acid levels, remained within normal range when supplementation was discontinued. Of the four patients that could discontinue use of VAL and ILE supplementation, one was also able to completely discontinue use of precursor-free medical food. The single patient who did not have VAL and ILE supplementation discontinued went on to have further diet changes made outside the study time frame, with an anticipated goal of safely stopping supplementation. Dietary adjustments for this patient have been delayed due to poor patient follow-up.

Discontinuation of amino acid supplementation, reduction of medical foods, and increase in natural protein meeting the RDAs for age and sex have several clear benefits. This management facilitates decreased reliance on free amino acids, reduced cost of dietary therapy, and a lighter burden on patients and/or caregivers for daily management of MMA, which in turn may improve quality of life [14]. In addition to depressing plasma levels of VAL and ILE, elevated intake of LEU has been associated with appetite suppression [6]. This phenomenon is widely seen in MMA, although it is unclear how much is related to high LEU levels, versus MMA itself, versus some other unknown factor. Further investigation is needed to assess whether the increased LEU intake from precursor-free medical food additionally impacts anorexia reported in MMA; however, future efforts to reduce LEU intake may have a significantly beneficial impact on patient appetite stimulation and overall patient nutritional well-being.

This study, in terms of design and size, has limitations. In our relatively large metabolic center, the number of patients with MMA due to MUT deficiency is low, particularly compared to the internationally acquired cohort at the NIH. Larger studies to continue to validate findings from Manoli et al. would be helpful. However, our cohort, even with an n of 5, was large enough to show impact and clinical practicality of the guidelines recommended. Given this was a retrospective study, certain markers were not reviewed due to lack of availability. The data extracted were solely focused on plasma amino acids, intake of protein and LEU from precursor-free medical food and natural protein, and use of VAL and/or ILE supplementation. Anthropometrics, nutritional laboratory monitoring (e.g., micronutrients, vitamins, and pre-albumin), and analysis of oral dietary intake, including total energy intake, were not reviewed or collected in this study. The authors acknowledge that this data would be beneficial, and may be used in future prospective research to assess impact of dietary changes on growth, laboratory values, and overall disease control in diet-treated organic acidemias. 

We have demonstrated the feasibility of decreasing use of precursor-free medical food, while continuing to meet (and better approximate) the RDA recommendations for protein and allowing for the discontinuation of individual amino acid supplementation. Additionally, based on the evidence presented in this study, when meeting or exceeding RDA recommendations for natural protein intake, it may not be necessary to provide any additional protein via precursor-free medical food in patients with MUT-deficient MMA. With these two changes, plasma propiogenic amino acids remained within reference range, demonstrating safety and diminishing the possibility of iatrogenic disturbances in amino acid metabolism.

## Figures and Tables

**Figure 1 nutrients-12-00473-f001:**
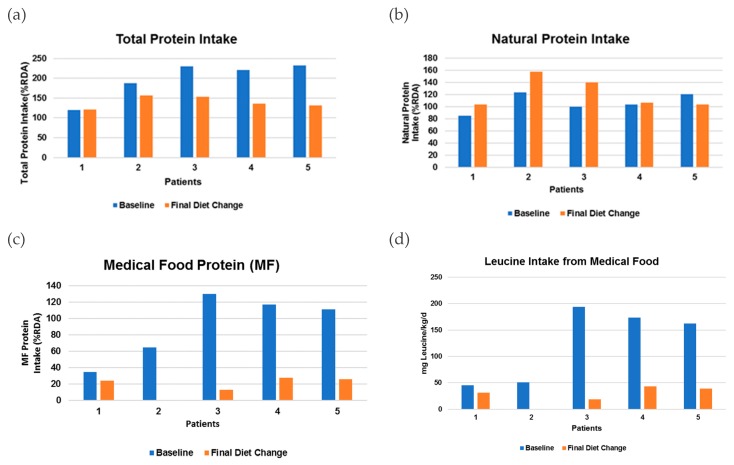
Changes in intake for each patient from baseline to final diet change. The Recommended Dietary Allowance (RDA) for protein intake was calculated based on the Institute of Medicine guidelines [7]. (**a**) Total protein intake from baseline to final diet change, *p* = 0.035; (**b**) natural protein intake from baseline to final diet change, *p*= 0.204; (**c**) medical food intake from baseline to final diet change, *p* = 0.014; and (**d**) leucine intake from medical food from baseline to final diet change, *p*= 0.0276.

**Table 1 nutrients-12-00473-t001:** Paired sample t-test results of dietary composition reflected as percentage of the Recommended Dietary Allowance (RDA).

Protein Intake	Baseline *n = 5*	Final Diet *n = 5*	*p*-Value*
Total Protein Intake	198% (41.8%)	141% (13.5%)	*p* = 0.035
Natural Protein Intake	107% (14.3%)	123% (22.3%)	*p* = 0.204
Protein Intake from Medical Food	92% (35.8%)	18% (10.4%)	*p* = 0.014

Data presented as the mean (SD). *p* < 0.05 *.

**Table 2 nutrients-12-00473-t002:** Baseline diet to final diet change in relation to total protein intake, natural protein intake, and protein intake from precursor-free medical food, *n* = 5 patients (pt).

Baseline Protein	Pt. 1	Pt. 2	Pt. 3	Pt. 4	Pt. 5
Baseline Total Protein	121%	188%	230%	221%	232%
Baseline Natural Protein	86%	123%	100%	104%	121%
Baseline Medical Food Protein	35%	65%	130%	117%	111%
Baseline Medical Food Leucine *	130%	115%	496%	417%	444%
Final Total Protein	127%	157%	153%	136%	131%
Final Natural Protein	103%	157%	140%	107%	104%
Final Medical Food Protein	24%	0%	13%	28%	26%
Final Medical Food Leucine *	0%	80%	47%	99%	111%

Data presented as the percentage of the RDA [7]. * Based on 2007 World Health Organization/Food and Agriculture Organization/United Nations University (WHO/FAO/UNU) Requirements for Leucine intake [8].

**Table 3 nutrients-12-00473-t003:** Paired sample t-test results of plasma propiogenic amino acids.

	Baseline *n = 5*	Final Diet *n = 5*	*p*-Value *	Reference Range **
Plasma Valine (µM)	109 (25.0)	92 (17.7)	*p* = 0.385	74–321
Plasma Isoleucine (µM)	38 (16.6)	27 (7.1)	*p* = 0.226	22–107
Plasma Leucine (µM)	128 (69.3)	63 (14.3)	*p* = 0.113	49–216
Plasma Methionine (µM)	20 (4.5)	23 (1.9)	*p* = 0.179	7–47
Plasma Threonine (µM)	93 (35.1)	148 (50.4)	*p* = 0.011	35–226

Data presented as the mean (SD). ** p < 0.05*, ** plasma amino acid reference ranges of the Children’s Hospital Colorado Biochemical Laboratory.

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
