# Peer review of "Impact on Isoleucine and Valine Supplementation When Decreasing Use of Medical Food in the Nutritional Management of Methylmalonic Acidemia"

_nutrients, 2020, doi:10.3390/nu12020473_

Round 1

Reviewer 1 Report

An interesting observations of the clinical practice. What was the followup of all included patients? How often plasma amino acids was reported?

Was the dose of 550 mg/day of ILE relevant for paediatric and adult patients? Where does it come from?

What was the clinical relevance of the clinical study? Was a muscle mass measured I.e. in body composition? Was the weight and height assessed? It may be clinically relevant to follow these patients up longterm to see the benefit of the AA supplementation.

Could the authors discuss it in more details? 

Reviewer 2 Report

I think that it is an interesting contribution about the new dietary approach of the MMA.

Reviewer 3 Report

Review: Impact on isoleucine and valine supplementation when decreasing use of medical food for the nutritional management of MMA.

This article describes the impact of changing dietary protein sources (primarily reducing use of medical food) for 5 patients with methylmalonyl mutase deficiency followed at Children’s Hospital Colorado. Although this is a small number of subjects, it is a number typical of the patient load of many metabolic centers. The statistical findings from this small number really highlights the impact of the recent reevaluation of historical guidelines for diet management of MMA. It was important to replicate the findings of Manoli et al who were able to evaluate a much larger study cohort. For metabolic dietitians, this study is a significant contribution for the treatment of patients with MMA.

In general, the paper includes all the necessary information, but I did find some areas the need further clarification:

Line 20 and 21 in the abstract was confusing. I suggest making this into two sentences. “ ….100% received ILE supplementation. After intervention, only one of the cohort….”

Starting with line 56: I suggest this: “….from the presentation: Methylmalonic academia…. presented by Irini Manoli MD, PhD from the Organic Acidemia (add name of NIH program) for the 5th GMDI Educational Conference and subsequently published……by Manoli et al. I don’t think including the name of the conference is important, but readers need to know where Dr. Manoli is from since this paper had a large impact on our review MMA management.

In Materials and Methods, line 87: I need more details about the “dietary changes”. From results, this was primarily a reduction in medical food rather than intact protein. I think this needs to be explained further. What was your endpoint to decide to stop making further diet changes – plasma amino acids? What were you striving for as your endpoint? To meet the RDA? Normalize leucine concentrations? Some additional details in Methods would be helpful to better understand what you were trying to achieve.

In Results, line 142. A couple sentences rather than just one would be this clearer. “At baseline, ratios were XXXXX. After the final diet change, average ratios decreased to XXXX.” Are there normal values for these ratios? If yes, it would be good to add this so readers know what to compare it to.

In Discussion, line 164. What is the mechanism for elevated LEU to reduce EAA concentrations? Is it reduced absorption with competitive inhibition at gut level? If mechanism isn’t known, probably want to say that.

In Discussion, end of line 169. Include reference numbers for the Manoli paper and any other paper that questions previous dietary therapy. There are several in your reference list that should be listed here.

Discussion, line 184: Suggest changing this to “…recommended by the RDA, plasma VAL and ILE concentrations suggested deficient intake and had to be supplemented”. They further suggest limiting the addition of medical food protein to meet a total protein intake of 100 to 120% of RDA”. As it is written, it reads as we need to limit medical food protein to meet 100 – 120% of the RDA, not the combination of intact and medical food protein to meet 100 to 120% RDA.

Discussion, line 205: “…caregivers for daily diet management, which in turn may…”. I would say that the diet changes make the entire diet easier rather than just the medical food portion.

Round 2

Reviewer 1 Report

The paper has significantly improved.